# Risk factors for 3-month mortality in bedridden patients with hospital-acquired pneumonia: A multicentre prospective study

Jing Jiao[1☯], Zhen Li[1☯], Xinjuan Wu[1]*, Jing Cao[1], Ge Liu[2], Ying Liu[1], Fangfang Li[3], Chen Zhu[1], Baoyun Song[4], Jingfen Jin[5], Yilan Liu[6], Xianxiu Wen[7], Shouzhen Cheng[8], Xia Wan[9]

1 Department of Nursing, Peking Union Medical College Hospital, Beijing, China, 2 Department of Surgery, Peking Union Medical College Hospital, Beijing, China, 3 Department of Medicine, Peking Union Medical College Hospital, Beijing, China, 4 Department of Nursing, Henan Provincial People's Hospital, Zhengzhou, Henan Province, China, 5 Department of Nursing, The Second Affiliated Hospital Zhejiang University School of Medicine, Hangzhou, Zhejiang Province, China, 6 Department of Nursing, Wuhan Union Hospital, Wuhan, Hubei Province, China, 7 Department of Nursing, Sichuan Provincial People's Hospital, Chengdu, Sichuan Province, China, 8 Department of Nursing, The First Affiliated Hospital, Sun Yat-sen University, Guangzhou, Guangdong Province, China, 9 Institute of Basic Medical Sciences, Chinese Academy of Medical Sciences and School of Basic Medicine, Peking Union Medical College, Beijing, China

☯ These authors contributed equally to this work.
* wuxinjuan@sina.com

**Data Availability Statement:** The Ethics Committee of Peking Union Medical College Hospital that approved the study prohibits the authors from making the research data set publicly

## Abstract

### Background

Mortality among patients with hospital-acquired pneumonia (HAP) is quite high; however, information on risk factors for short-term mortality in this population remains limited. The aim of the current study was to identify the risk factors for mortality in bedridden patients with HAP during a 3-month observation period.

### Methods

A secondary data analysis was conducted. In total, 1141 HAP cases from 25 hospitals were included in the analysis. Univariate and multilevel regression analyses were performed to identify the risk factors for mortality.

### Results

During the 3-month observation period, there were 189 deaths among bedridden patients with HAP. The mortality rate in this study was 16.56%. Multilevel regression analysis showed that ventilator-associated pneumonia (OR = 2.034, 95%CI: 1.256, 3.296, p = 0.004), pressure injuries (OR = 2.202, 95%CI: 1.258, 3.852, p = 0.006), number of comorbidities (OR = 1.076, 95%CI: 1.016,1.140, p = 0.013) and adjusted Charlson Comorbidity Index score (OR = 1.210, 95%CI: 1.090, 1.343, p<0.001) were associated with an increased risk of mortality, while undergoing surgery with general anaesthesia (OR = 0.582, 95%CI: 0.368, 0.920, p = 0.021) was associated with a decreased risk of mortality.

available. Access to data may be requested via the Ethics Committee of Peking Union Medical College Hospital (pumchkyc@126.com).

**Funding:** This work was supported by the Special Scientific Research Fund of Public Welfare Profession of China [grant number 201502017]. The funders had no role in the study design, data collection and analysis, decision to publish, or preparation of the manuscript.

**Competing interests:** The authors have declared that no competing interests exist.

## Conclusions

The identification of risk factors associated with mortality is an important step towards individualizing care plans. Our findings may help healthcare workers select high-risk patients for specific interventions. Further study is needed to explore whether appropriate interventions against modifiable risk factors, such as reduced immobility complications or ventilator-associated pneumonia, could improve the prognoses.

## Introduction

Hospital-acquired pneumonia (HAP), or nosocomial pneumonia, is a pulmonary infection that develops in patients hospitalized for more than 48 hours in the intensive care unit (ICU) or in other wards [1].

Ventilator-associated pneumonia (VAP) is a subcategory of HAP that occurs in mechanically ventilated patients more than 48 hours after tracheal intubation [2]. HAP is one of the most common hospital-acquired infections [3]. In the United States, HAP is the second most common nosocomial infection, with an incidence ranging from 5 to more than 20 cases per 1000 hospital admissions [4]. In China, the incidence of hospital-acquired infections in all hospitalized patients was 3.22–5.22%, among which hospital-acquired lower respiratory tract infections accounted for 1.76–1.94% [5]. HAP/VAP account for the largest proportion of hospital-acquired infections worldwide.

HAP can have negative consequences for patients, including prolonged hospital stay, decreased quality of life and high mortality [6,7]. Despite improvements in prevention, antimicrobial therapy, and supportive care [1], HAP remains an important cause of morbidity and mortality [8]. The mortality rate for HAP ranges from 38% to more than 70% [9]. In Japan, the 30-day mortality was 23.8% [10]. A recent review found that the incidence of VAP ranges from two to 16 episodes per 1000 ventilator-days, with an attributable mortality rate of 3–17% [11].

Compared to a large number of studies on the risk factors for mortality among community acquired pneumonia (CAP) patients [12,13], only a relatively small number of studies have focused on HAP. Feng et al found that the risk factors included age >70 years, ICU admission, blood lymphocyte count, multidrug-resistant gram-negative bacteria (MDR-GNB) infection, and blood urea nitrogen (BUN) level [14]. Another study in a psychiatric hospital found that the mortality rate in HAP patients tended to increase with increasing severity classification [15]. Another recent study found that comorbidities positively contribute to the risk of death from pneumonia in the hospital; however, the study did not distinguish their types or origins [16]. The specific factors associated with HAP remain unknown.

## Materials and methods

### The aim, design and setting of the study

The current study aimed to: (i) describe the mortality rate of bedridden patients with HAP and (ii) explore the risk factors for mortality among the study population.

The data for this study were obtained from a database of multicentre baseline survey data from a national research programme that aimed to explore and construct a standardized nursing care model for immobility complications (including pressure injuries, pneumonia, deep vein thrombosis, and urinary tract infection) among bedridden inpatients [17]. The details of

the programme have been described elsewhere [17–19]. In brief, the study was carried out in 25 hospitals (six tertiary, 12 non-tertiary and seven community hospitals) in northern China (Beijing), central China (Henan and Hubei), southern China (Guangdong), eastern China (Zhejiang), and western China (Sichuan) from November 2015 to June 2016. Participants were recruited from wards with high proportions of bedridden patients, including neurology, neurosurgery, general medicine, orthopaedics, geriatrics and critical care units.

## Study population

Patients were eligible for the study if they were: (a) adult patients (≥18 years old); (b) bedridden for at least one day after admission in the selected ward (bedridden was defined as the patients' basic physiological needs being carried out in bed except for active or passive bedside standing/wheelchair use for examination or treatment) [20]; and (c) diagnosed with HAP. Patients were excluded if they had more than one type of major complication of immobility (pressure injuries, deep vein thrombosis, urinary tract infection) at the time of enrollment. Overall, 1141 patients were included in the analysis.

## Measures

**HAP.** HAP was defined as pneumonia occurring at least 48 hours after hospital admission, excluding any infection present at the time of admission [5]. It was diagnosed based on a combination of clinical, laboratory and radiological data by physicians and was recorded in the patient's medical record. The criteria for the diagnosis of HAP included new pulmonary infiltrate confirmed by X-ray or computed tomography (occurring ≥48 hours after admission) associated with at least one of the following: new or worsening cough with or without sputum production, fever (temperature >37.8 ˚C) or hypothermia (<35.6 ˚C), leucocytosis, left shift, or leucopenia based on local normal values [5].

**Sociodemographic and disease characteristics.** Patients' sociodemographic characteristics (such as age, gender and health insurance) were self-reported by the patients. Mobility at admission, pressure injuries, comorbidities and other medical data information were obtained from audits of the patients' hospital charts and electronic health records. The age-adjusted Charlson Comorbidity Index (ACCI) was calculated based on the diagnosis at discharge [21]. International Classification of Diseases, 10th Edition (ICD-10) codes were used for the diagnosis code and calculation.

**Outcomes.** In this study, 3-month all-cause mortality was evaluated as the primary outcome. Mortality information was observed and recorded for 90 days after the enrollment date regardless of whether the patient died before or after discharge.

## Data collection

Eligible participants were identified and enrolled by pre-trained nurses in the selected wards. In total, 787 registered nurses from 25 general hospitals were appointed to perform patient recruitment and data collection. They screened and recorded the participating patients' information daily. An electronic data collection (EDC) system was designed for data collection for this study. The evaluation of patients and demographic data were collected on the day of recruitment. Diagnosis of HAP and other related variables were collected on a daily basis during the hospitalization period. After discharge, participants received a telephone follow-up call every two weeks until their 90[th] day after enrollment. A strictly standardized procedure were applied to ensure the quality of data collection. A coordinator in each hospital was responsible for internal logistics and checking compliance to the study protocol. The head nurses of the selected wards regularly audited the data recorded by the data collectors. Moreover, an

independent quality control team regularly supervised the data collection through on-site quality control and case verification measures and provided feedback via emails, phones, and meetings.

### Ethical considerations

The study was approved by the Ethics Committee of Peking Union Medical College Hospital (S-700). Patients received verbal and written informed consent. If patients had a cognitive impairment (such as sedation), their family members was asked to provide the consent. All data were anonymous or were kept confidential.

### Data analysis

Continuous and categorical variables are presented as the means (standard deviations (SDs)) and n (%), respectively. The variables were organized into three levels: hospital, ward, and patient. Intra-class coefficients (ICCs) were calculated to gauge the potential effect of clustering on the results. We used the Mann-Whitney U test, $\chi^2$ test, or Fisher's exact test to compare differences between survivors and non-survivors where appropriate. To explore the risk factors associated with mortality, univariate and multivariate analyses were conducted using a multilevel mixed-effects model with two random intercept models to evaluate the effect of the hospital and ward. The model considered that patients from the same hospital or ward may have unmeasured characteristics that affect outcomes associated with those hospital or wards differently than those associated with different hospitals or wards. Death status was coded as the number 0 or 1, with 0 indicating survival and 1 indicating death. Covariates that were considered to be important factors based on prior literature and the univariate analysis were considered candidates for inclusion in the models. Hospital-level variables included hospital type (tertiary and non-tertiary hospitals), and patient-level variables included age, gender, education level, health insurance, smoking status, body mass index (BMI), mobility at admission, surgery with general anaesthesia, VAP, length confined to bed, pressure injuries and comorbidities. The statistical analyses were carried out using SAS 9.4 for Windows (SAS Institute, Cary, North Carolina, USA). All tests were 2-tailed, with $p<0.05$ considered significant.

## Results

Among 1141 hospitalized bedridden patients with HAP, 189 deaths occurred during the 3-month follow-up period, leading to an overall mortality rate of 16.56%.

Demographic characteristics and the main clinical features of the patient groups are compared in Table 1. Compared to survivors, non-survivors tended to be older, have a lower BMI, and did not undergo surgery with general anaesthesia; additionally, they were more likely to be admitted to a non-tertiary hospital and have dependent mobility at admission. They also had more comorbidities and a higher age-adjusted Charlson Comorbidity Index score than survivors.

Table 2 shows the results of the univariate and multivariate multilevel mixed-effects model analysis after adjustment for a series of covariates. In the multilevel model, we found that VAP (OR = 2.034, 95%CI: 1.256, 3.296, p = 0.004), pressure injuries (OR = 2.202, 95%CI: 1.258, 3.852, p = 0.006), number of comorbidities (OR = 1.076, 95%CI: 1.016,1.140, p = 0.013) and adjusted Charlson Comorbidity Index score (OR = 1.210, 95%CI: 1.090, 1.343, p<0.001) were associated with an increased risk of mortality, while undergoing surgery with general anaesthesia (OR = 0.582, 95%CI: 0.368, 0.920, p = 0.021) was associated with a decreased risk of mortality.

**Table 1. Baseline characteristics of bedridden patients with hospital-acquired pneumonia (n = 1,141).**

| Variables | Total | Non-survivor (n = 189) | Survivor (n = 952) | t/$\chi^2$/Z | p value |
|---|---|---|---|---|---|
| **Gender** | | | | | |
| Male | 695(60.91) | 122(64.55) | 573(60.19) | 1.260 | 0.2617 |
| Female | 446(39.09) | 67(35.45) | 379(39.81) | | |
| **Age, years** | 62.80±16.58 | 70.38±15.02 | 61.30±16.47 | 7.032 | <0.0001 |
| **Age group** | | | | | |
| 18~64 | 575(50.39) | 54(28.57) | 521(54.73) | 58.023 | <0.0001 |
| 65~74 | 247(21.65) | 42(22.22) | 205(21.53) | | |
| ≥75 | 319(27.96) | 93(49.21) | 226(23.74) | | |
| **Highest education level** | | | | | |
| Primary school or below | 520(45.57) | 97(51.32) | 423(44.43) | 3.770 | 0.1518 |
| Junior or senior high school | 488(42.77) | 69(36.51) | 419(44.01) | | |
| College degree or above | 133(11.66) | 23(12.17) | 110(11.55) | | |
| **Health insurance** | | | | | |
| No | 277(19.89) | 32(16.93) | 195(20.48) | 1.248 | 0.2639 |
| Yes | 914(80.11) | 157(83.07) | 757(79.52) | | |
| **Smoking status** | | | | | |
| No | 847(74.23) | 148(78.31) | 699(73.42) | 1.965 | 0.1609 |
| Yes | 294(25.77) | 41(21.69) | 253(26.58) | | |
| **Body Mass Index (BMI)** | | | | | |
| <18.5 | 97(8.51) | 29(15.34) | 68(7.15) | 15.489 | 0.0004 |
| 18.5~23.9 | 592(51.93) | 99(52.38) | 493(51.84) | | |
| >23.9 | 451(39.56) | 61(32.28) | 390(41.01) | | |
| **Mobility at admission** | | | | | |
| Independent | 277(24.28) | 27(14.29) | 250(26.26) | 12.301 | <0.001 |
| Dependent | 864(75.72) | 162(85.71) | 702(73.74) | | |
| **Surgery with general anaesthesia** | | | | | |
| No | 733(64.24) | 150(79.37) | 583(61.24) | 22.553 | <0.001 |
| Yes | 408(35.76) | 39(20.63) | 369(38.76) | | |
| **VAP** | | | | | |
| **No** | 957(83.87) | 150(79.37) | 807(84.77) | 3.405 | 0.0650 |
| **Yes** | 184(16.13) | 39(20.63) | 145(15.23) | | |
| **Length of confinement to bed** | | | | | |
| 1~7 | 232(20.33) | 31(16.40) | 201(21.11) | 3.829 | 0.1474 |
| 8~14 | 317(27.78) | 62(32.80) | 255(26.79) | | |
| ≥15 | 592(51.88) | 96(50.79) | 496(52.10) | | |
| **Hospital type** | | | | | |
| Non tertiary | 205(17.97) | 54(28.57) | 151(15.86) | 17.284 | <0.001 |
| Tertiary | 936(82.03) | 135(71.43) | 801(84.14) | | |
| **Pressure injuries** | | | | | |
| No | 1041(91.24) | 160(84.66) | 881(92.54) | 12.264 | <0.001 |
| Yes | 100(8.76) | 29(15.34) | 71(7.46) | | |
| **Comorbidities (n = 1,135)** | | | | | |
| Number of comorbidities (IQR) | 5(5) | 7(6) | 5(5) | 5.8922 | < .0001 |
| 1–4 | 480(43.52) | 48(26.37) | 432(46.91) | 26.064 | <0.001 |
| ≥5 | 623(56.48) | 134(73.63) | 489(53.09) | | |
| ACCI score, median (IQR) | 3(4) | 5(3) | 3(3) | 8.391 | <0.001 |
| Mild (ACCI 0–2) | 489(43.08) | 38(20.11) | 451(47.67) | 75.398 | <0.001 |

*(Continued)*

**Table 1.** (Continued)

| Variables | Total | Non-survivor (n = 189) | Survivor (n = 952) | t/χ²/Z | p value |
|---|---|---|---|---|---|
| Moderate (ACCI 3–4) | 314(27.67) | 49(25.93) | 265(28.01) | | |
| Severe (ACCI≥5) | 332(29.25) | 102(53.97) | 230(24.31) | | |

Note: IQR: Interquartile range; VAP: Ventilator-associated pneumonia; ACCI: Age-adjusted Charlson Comorbidity Index.

## Discussion

This multicentre prospective study identified a relatively high mortality rate as well as several risk factors for mortality in bedridden inpatients with HAP. In particular, VAP, pressure injuries, number of comorbidities, and adjusted Charlson Comorbidity Index score were associated with increased risk of death, while undergoing surgery with general anaesthesia was associated with a decreased risk of mortality.

Despite advances in new guidelines and significant efforts to improve the care and outcomes associated with HAP, we found that the overall 3-month mortality among hospitalized bedridden patients with HAP remains high (16.56%), which is in concordance with other previous studies [22,23]. A study conducted in the United States found that mortality in adults hospitalized with pneumonia was 13.0%, 23.4%, and 30.6% at 30 days, 6 months, and 1 year, respectively [23]. The results highlight the importance of taking specific measures to improve the prognosis of bedridden patients with HAP.

In the multivariate multilevel analysis, we found several risk factors for mortality among bedridden patients with HAP, among which pressure injuries had the highest odds ratio. A meta-analysis found that patients complicated with pressure injuries are estimated to have a two times higher risk of mortality compared to patients without pressure injuries [24]. Immobility complications, including pressure injuries, have been shown to be predictors of adverse outcomes and prognosis among bedridden patients [17]. Another study found that the

**Table 2. Risk factors associated with death among bedridden patients with hospital-acquired pneumonia (n = 1,141).**

| Variables | Univariate OR (95% CI) | p value | Multivariable OR (95% CI) | P value |
|---|---|---|---|---|
| Female (vs male) | 0.823(0.592, 1.144) | 0.247 | 0.729(0.494, 1.076) | 0.111 |
| Age, years | 1.039(1.027, 1.051) | <0.001 | 1.005(0.989, 1.021) | 0.580 |
| College degree or above (vs primary school or below) | 0.983(0.565, 1.710) | 0.950 | 0.965(0.516, 1.807) | 0.998 |
| Junior or senior high school (vs primary school or below) | 0.760(0.522, 1.106) | 0.144 | 0.828(0.543, 1.264) | 0.365 |
| Health insurance (vs no insurance) | 1.219(0.794, 1.871) | 0.366 | 1.076(0.676, 1.712) | 0.757 |
| Current smoker (vs nonsmoker) | 1.258(0.530, 1.129) | 0.183 | 0.705(0.448, 1.109) | 0.131 |
| BMI<18.5 (vs BMI 18.5~23.9) | 2.148(1.287, 3.583) | 0.005 | 1.519(0.862, 2.679) | 0.143 |
| BMI >23.9 (vs BMI 18.5~23.9) | 0.771(0.535, 1.110) | 0.155 | 0.807(0.543, 1.198) | 0.278 |
| Dependent mobility at admission (vs independent mobility) | 2.265(1.446, 3.440) | <0.001 | 1.467(0.882, 2.440) | 0.140 |
| Surgery with general anaesthesia (vs no surgery) | 0.383(0.259, 0.569) | <0.001 | 0.582(0.368, 0.920) | 0.021 |
| VAP (vs none VAP) | 1.483(0.990, 2.223) | 0.056 | 2.034(1.256, 3.296) | 0.004 |
| Length of confinement to bed | 1.102(0.896, 1.357) | 0.357 | 0.905(0.713, 1.148) | 0.410 |
| Admission to a tertiary hospital (vs nontertiary hospital) | 0.469(0.315, 0.687) | <0.001 | 0.625(0.358, 1.090) | 0.093 |
| Pressure injuries (vs no pressure injury) | 2.565(1.565, 4.201) | <0.001 | 2.202(1.258, 3.852) | 0.006 |
| Number of comorbidities | 1.168(1.112, 1.228) | <0.001 | 1.076(1.016, 1.140) | 0.013 |
| ACCI score | 1.327(1.238, 1.423) | <0.001 | 1.210(1.090, 1.343) | <0.001 |

Note: CI, confidence interval; VAP: Ventilator-associated pneumonia; AACI, age-adjusted Charlson Comorbidity Index.

mortality of patients with pressure injuries was as high as 66% in a 12-week median follow-up period [25]. Our study further indicated that pressure injuries are a good predictor of death. As this factor is modifiable, particular attention should be given to the prevention and treatment of pressure injuries and other immobility complications among the population.

Mortality was significantly higher in patients with VAP. Similar to our study, VAP was the leading cause of death in mechanically ventilated patients [26]. A recent study in the Chinese population found that the 30-day mortality rate of VAP patients was 42.8% [27]. A systematic review indicated that the attributable mortality of ventilator-associated pneumonia is mainly caused by increased length of ICU stay [28]. Together with the findings of the current study, this suggests that healthcare providers should pay more attention to patients with VAP to reduce their mortality.

The number of comorbidities was significantly associated with short-term mortality in our study. A previous study found that comorbidities positively contributed to the risk of death from pneumonia in the hospital, regardless of their type or origin [16]. Another study found an inconsistent result indicating that the number of comorbidities was not associated with mortality [29]. Reasons for the difference may be that, in that study, the study population was elderly (mean age: 81.6 years), and the majority had multiple comorbidities; few patients were aged <65 years or had few comorbidities. The participants in the current study were much younger (mean age: 62.80 years). An even distribution of the number of diseases simplifies the process of detecting the impact.

With regard to the severity of illness, previous studies have mainly focused on Pneumonia Severity Index (PSI) and CURB-65 score, which were mainly used to predict mortality in patients with CAP [30] and were seldom trained or designed in patients with HAP [31]. The lack of severity assessment tools specifically for this group of patients has limited the ability to early detection of poor prognosis among HAP patients. The current study added new evidence by exploring the role of chronic comorbidities in predicting HAP patients' outcomes. We found that the ACCI score was an independent risk factor for mortality among HAP patients. The ACCI is widely used to predict mortality and has been validated in various clinical populations [21,32]. The results support the prognostic value of the ACCI score and indicate that the ACCI score may be useful for risk stratification and decision making of individual HAP patients. Future studies are necessary to compare its performance with CURB-65 and PSI and determine if this tool may be universally applied to HAP patients.

In addition to the above results, it noteworthy that patients who underwent surgery with general anaesthesia had a lower risk for mortality than those who did not. In the current study, 35.76% of patients underwent surgery with general anaesthesia. The reason might be that one of the most important surgical indications is a relatively good health condition.

## Strengths and limitations

The current study provides detailed mortality information of bedridden patients with HAP over a 3-month period. The major strengths of this study were the multicentre design and large population of consecutively included and prospectively followed patients, which increased the generalizability of the results. However, there are still several limitations. First, this was a secondary data analysis; thus, there is a lack of other relevant clinical data, which might have increased the possibility of information bias and misclassification. Second, adjustments were made for demographic, clinical and microbiological variables. However, some important patient characteristics (pneumonia severity, co-infections, and types of bacteria) could not be analysed because they were not available in the database. Prospective studies with more specific evaluations are required to confirm the findings. Third, the present study used

3-month follow-up data, and the long-term risk factors for mortality therefore could not be determined. Future studies should extend the follow-up period and include large-scale related factors.

## Conclusions

In the current study, we identified several factors (VAP, pressure injuries, numbers of comorbidities, ACCI and surgery with general anaesthesia) for mortality among bedridden HAP patients. These findings have great clinical significance for improving the prognosis of patients. Early identification and intervention for these patients, who are at high risk of mortality, will help to improve clinical outcomes. We recommend further research on whether the modification of factors associated with increased mortality, such as improving care quality to prevent immobility complications, could improve outcomes in bedridden HAP patients.

## Author Contributions

**Conceptualization:** Xinjuan Wu.

**Formal analysis:** Zhen Li.

**Funding acquisition:** Xinjuan Wu.

**Investigation:** Zhen Li, Jing Cao, Ge Liu, Ying Liu, Fangfang Li, Chen Zhu, Baoyun Song, Jingfen Jin, Yilan Liu, Xianxiu Wen, Shouzhen Cheng, Xia Wan.

**Methodology:** Jing Jiao.

**Writing – original draft:** Jing Jiao, Zhen Li.

**Writing – review & editing:** Jing Jiao.

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
