## [Decision Letter · Decision Letter 0]

11 Dec 2020

PONE-D-20-25311

Risk factors for 3 - month mortality in bedridden patients with hospital - acquired pneumonia: a multicenter prospective study.

PLOS ONE

Dear Dr. Li,

Thank you for submitting your manuscript to PLOS ONE. After careful consideration, we feel that it has merit but does not fully meet PLOS ONE’s publication criteria as it currently stands. Therefore, we invite you to submit a revised version of the manuscript that addresses the points raised during the review process.

We look forward to receiving your revised manuscript.

Kind regards,

Shane Patman, PhD

Academic Editor

PLOS ONE

Journal Requirements:

2.) We suggest you thoroughly copyedit your manuscript for language usage, spelling, and grammar. If you do not know anyone who can help you do this, you may wish to consider employing a professional scientific editing service.  

3.) Thank you for stating in your financial disclosure: 

'The funders had no role in study design, data collection and analysis, decision to publish, or preparation of the manuscript.'

4.) We note that you have indicated that data from this study are available upon request. PLOS only allows data to be available upon request if there are legal or ethical restrictions on sharing data publicly. For information on unacceptable data access restrictions, please see http://journals.plos.org/plosone/s/data-availability#loc-unacceptable-data-access-restrictions.

5.) Your ethics statement should only appear in the Methods section of your manuscript. If your ethics statement is written in any section besides the Methods, please delete it from any other section.

6.) We noticed you have some occurrence of overlapping text with the following previous publication(s), which needs to be addressed:

https://journals.plos.org/plosone/article?id=10.1371%2Fjournal.pone.0205729

In your revision ensure you cite all your sources (including your own works), and quote or rephrase any duplicated text outside the methods section. Further consideration is dependent on these concerns being addressed.

Reviewers' comments:

Reviewer's Responses to Questions

**Comments to the Author**

1. Is the manuscript technically sound, and do the data support the conclusions?

Reviewer #1: Yes

Reviewer #2: Partly

2. Has the statistical analysis been performed appropriately and rigorously? 

Reviewer #1: Yes

Reviewer #2: I Don't Know

3. Have the authors made all data underlying the findings in their manuscript fully available?

Reviewer #1: Yes

Reviewer #2: Yes

4. Is the manuscript presented in an intelligible fashion and written in standard English?

Reviewer #1: Yes

Reviewer #2: Yes

5. Review Comments to the Author

Reviewer #1: The authors identified the risk factors for mortality in bedridden patients with HAP during a 3-month observation period. They found that intensive care unit (ICU) admission, number of comorbidities and adjusted Charlson Comorbidity Index score were associated with an increased risk of mortality. However, several previous studies have shown similar results that disease severity and presence of comorbidities are associated with poor prognosis. The authors should clarify new evidences of this study.

Reviewer #2: In the manuscript “3-month mortality in bedridden patients with hospital-acquired pneumonia; a multicenter prospective study”, Jiao et al report the results from a secondary analysis of a prospective observational study that included 1141 patients diagnosed with HAP from 25 hospitals. They describe associations between patient characteristics, ward status, and treatments factors (surgical and anesthesia) versus inpatient mortality. They report ICU admission and patient comorbidities to be positively associated with death, and surgery with GA and tertiary hospital admission to be negatively associated with death, and conclude that the findings could help providers identify patients with high.

The etiology and consequences of HAP are poorly characterized, and a large prospective study of patients with a standard sampling method has much potential for adding knowledge. However, as currently written, the analysis seems to fall short of advancing our understanding dramatically, and the conclusion that using the risk factors for decision-making does not seem to be supported by the results. Some makor concerns:

Sampling method: While the case definition of a bedridden patient and HAP was well described, the process of enrollment into the prospective cohort was not immediately clear. Were patients identified and enrolled by treating clinicians? Was there a screening process applied to all hospitalized patients? If the latter, how was this process applied? The authors mention that patients with major complications of immobility at the time of enrollment were excluded; what were these exlclusion criteria? What was the enrollment % of patients initially screened?

Treatments during the hospitalization including ventilator use, antibiotic use, and glucocorticoid use were listed as patient charcteristics. I do not typically view these factors as patient characteristics, as they are interventions that might be associated with the other patient characteristics and the outcomes in very complex ways. If the study is intended to be a cohort study and these interventions are being examined as potential independent risk factors for HAP, then the timing is also incredibly important and I was unable to discern at what point in the hospitalization – before or after the development of HAP – these interventioned were identified. The results would have very different meaning to me depending on this.

Conclusions: It is hard to glean what meaning the authors feel this study has to our understanding from the conclusion. It is not surpirising that comorbidities are associated with higher mortality; however, whether the aCCI score has utility over clinical judgment is not supported in this study. It is als not surpiring that patients in the ICU have higher mortlity risk, but the authors’ causal explanations when attempting to explain some associations (ie, ICU admission leading to ventilator use leading to mortality, or surgical admission � differences in quality of medical care � lower mortality) seem very far-reaching. It all is further muddied by an unclear picture of how these patients were identified, and how that identification process accounted for – or did not – the potential sampling bias that renders the findings difficult to generalize.

6. PLOS authors have the option to publish the peer review history of their article (what does this mean?). If published, this will include your full peer review and any attached files.

Reviewer #1: **Yes: **Mutsuo Yamaya

Reviewer #2: No

---

## [Author Response · Author response to Decision Letter 0]

22 Dec 2020

Dear Shane Patman,

Thank you for your letter and for the reviewers’ comments concerning our manuscript entitled “Risk factors for 3-month mortality in bedridden patients with hospital-acquired pneumonia: a multicentre prospective study” (ID: PONE-D-20-25311). Those comments were valuable and helpful for revising and improving the manuscript. We have taken the comments into careful consideration and have made corrections that we hope will be met with approval. Revised portions are marked in red in the paper. The main correction and the response to the reviewers’ comments are set out as below. We look forward to hearing from you regarding our submission. We would be glad to respond to any further questions and comments that you may have.

Journal Requirements:

Author response: Thank you for the reminder. We have revised the documents according to the journal’s style requirements.

2.) We suggest you thoroughly copyedit your manuscript for language usage, spelling, and grammar. If you do not know anyone who can help you do this, you may wish to consider employing a professional scientific editing service. 

Author response: Thank you for the reminder. The revised manuscript has been edited by AJE editors. 

3.) Thank you for stating in your financial disclosure: 

'The funders had no role in study design, data collection and analysis, decision to publish, or preparation of the manuscript.'

a. Please clarify the sources of funding (financial or material support) for your study. List the grants or organizations that supported your study, including funding received from your institution.

d. If you did not receive any funding for this study, please state: “The authors received no specific funding for this work.”

Author response: Thank you for the suggestion. We have included amended statements in the cover letter.

4.) We note that you have indicated that data from this study are available upon request. PLOS only allows data to be available upon request if there are legal or ethical restrictions on sharing data publicly. For information on unacceptable data access restrictions, please see http://journals.plos.org/plosone/s/data-availability#loc-unacceptable-data-access-restrictions.

Author response: Thank you for the reminder. We have added the statement in the updated cover letter. 

5.) Your ethics statement should only appear in the Methods section of your manuscript. If your ethics statement is written in any section besides the Methods, please delete it from any other section.

Author response: Thank you for the reminder. We have confirmed that the ethics statement only appears in the Methods section.

6.) We noticed you have some occurrence of overlapping text with the following previous publication(s), which needs to be addressed:

https://journals.plos.org/plosone/article?id=10.1371%2Fjournal.pone.0205729

In your revision ensure you cite all your sources (including your own works), and quote or rephrase any duplicated text outside the methods section. Further consideration is dependent on these concerns being addressed.

Author response: Thank you for your important comments. We have carefully cited the references and revised the text thoroughly in the current study. 

Reviewers' comments:

Reviewer's Responses to Questions

Comments to the Author

1. Is the manuscript technically sound, and do the data support the conclusions?

Reviewer #1: Yes

Reviewer #2: Partly

 2. Has the statistical analysis been performed appropriately and rigorously? 

 Reviewer #1: Yes

Reviewer #2: I Don't Know

3. Have the authors made all data underlying the findings in their manuscript fully available?

Reviewer #1: Yes

Reviewer #2: Yes

 4. Is the manuscript presented in an intelligible fashion and written in standard English?

Reviewer #1: Yes

Reviewer #2: Yes

5. Review Comments to the Author

Reviewer #1: The authors identified the risk factors for mortality in bedridden patients with HAP during a 3-month observation period. They found that intensive care unit (ICU) admission, number of comorbidities and adjusted Charlson Comorbidity Index score were associated with an increased risk of mortality. However, several previous studies have shown similar results that disease severity and presence of comorbidities are associated with poor prognosis. The authors should clarify new evidences of this study.

Author response: Thank you for your constructive and insightful advice about the findings. We agree that more evidence is needed to further clarify the study. We have included more patient-specific risk factors (including mobility at admission, immobility complications) and re-analysed the data. In the updated results, we additionally found that pressure injuries and VAP were risk factors for mortality among the population. The findings have great clinical implication for improving the prognosis of patients as they are modifiable factors. The detailed explanation and analysis of the results are described in the Discussion and Conclusion sections.

Reviewer #2: In the manuscript “3-month mortality in bedridden patients with hospital-acquired pneumonia; a multicenter prospective study”, Jiao et al report the results from a secondary analysis of a prospective observational study that included 1141 patients diagnosed with HAP from 25 hospitals. They describe associations between patient characteristics, ward status, and treatments factors (surgical and anesthesia) versus inpatient mortality. They report ICU admission and patient comorbidities to be positively associated with death, and surgery with GA and tertiary hospital admission to be negatively associated with death, and conclude that the findings could help providers identify patients with high.

The etiology and consequences of HAP are poorly characterized, and a large prospective study of patients with a standard sampling method has much potential for adding knowledge. However, as currently written, the analysis seems to fall short of advancing our understanding dramatically, and the conclusion that using the risk factors for decision-making does not seem to be supported by the results. Some makor concerns:

Sampling method: While the case definition of a bedridden patient and HAP was well described, the process of enrollment into the prospective cohort was not immediately clear. Were patients identified and enrolled by treating clinicians? Was there a screening process applied to all hospitalized patients? If the latter, how was this process applied? The authors mention that patients with major complications of immobility at the time of enrollment were excluded; what were these exlclusion criteria? What was the enrollment % of patients initially screened?

Author response: Thank you very much for your constructive advice about the sampling method. We apologize that this part was not described clearly in the original manuscript. We have revised the contents of this part of the manuscript (Page 3, line 140-151). The specific response to the questions you mentioned are listed as follows.

(1) The patients were identified and enrolled by pre-trained nurses. The screening process was only applied to bedridden patients (Page 3, line 140-144).

(2) The exclusion criteria were patients with more than one type of major complication of immobility (pressures injuries, deep vein thrombosis, and urinary tract infection) at the time of enrollment (Page 3, line 115-117). 

(3) Regarding the enrollment percentage, we agree that it is an important indicator to the representativeness of the sample and have attempted to acquire the percentage. However, we failed because this was a secondary data analysis. We have mentioned this in the limitation section (Page 12, line 265-271).

Treatments during the hospitalization including ventilator use, antibiotic use, and glucocorticoid use were listed as patient charcteristics. I do not typically view these factors as patient characteristics, as they are interventions that might be associated with the other patient characteristics and the outcomes in very complex ways. If the study is intended to be a cohort study and these interventions are being examined as potential independent risk factors for HAP, then the timing is also incredibly important and I was unable to discern at what point in the hospitalization – before or after the development of HAP – these interventioned were identified. The results would have very different meaning to me depending on this.

Author response: Thank you very much for your careful review and constructive suggestions with regard to the risk factors. I am also grateful for the specific modification strategy. We agree that the treatment during hospitalization should not be viewed as patient characteristics, as they were affected by patients’ characteristics and may have impacted the outcomes in complicated ways. I have made comprehensive and detailed changes according to your suggestion. On one hand, in the risk factor candidates, we excluded the factors related to treatment, including ICU admission, ventilator use, antibiotic use, and glucocorticoid use. On the other hand, we further took more patient-specific factors into consideration, including mobility at admission and immobility complications (pressure injuries). We reanalysed the data and updated the results and discussions.

Conclusions: It is hard to glean what meaning the authors feel this study has to our understanding from the conclusion. It is not surpirising that comorbidities are associated with higher mortality; however, whether the aCCI score has utility over clinical judgment is not supported in this study. It is als not surpiring that patients in the ICU have higher mortlity risk, but the authors’ causal explanations when attempting to explain some associations (ie, ICU admission leading to ventilator use leading to mortality, or surgical admission � differences in quality of medical care � lower mortality) seem very far-reaching. It all is further muddied by an unclear picture of how these patients were identified, and how that identification process accounted for – or did not – the potential sampling bias that renders the findings difficult to generalize.

Author response: Thank you for your thoughtful comments. We agree that it is essential to make more specific and reasonable analyses and explanations on the relationship between the risk factors identified and mortality. 

(1) In regard to comorbidities, previous studies on the comorbidities and mortality among pneumonia patients have mainly focused on PSI and CURB-65, both of which primarily use acute clinical findings to predict mortality. However, the role of chronic comorbidities in predicting outcomes is unclear. Therefore, we explored the ability of the ACCI to predict mortality in patients with HAP. We admit that more studies are needed on the performance and feasibility of ACCI. We have revised the discussion (Page 11, line 239-247).

(2) Regarding ICU admission, we excluded the variable in the multivariate analysis. As the reviewer suggested, it could not be a very good indicator to explain the mortality of HAP patients. In addition, ICU admission is also impacted by the patients’ decision making and total number of hospital ICU beds. In the updated results, we found that VAP increased the risk of mortality compared to non VAP patients. We have revised the discussion (Page 11, line 225-230).

 6. PLOS authors have the option to publish the peer review history of their article (what does this mean?). If published, this will include your full peer review and any attached files.

Do you want your identity to be public for this peer review? For information about this choice, including consent withdrawal, please see our Privacy Policy.

Reviewer #1: Yes: Mutsuo Yamaya

Reviewer #2: No

---

## [Decision Letter · Decision Letter 1]

9 Mar 2021

PONE-D-20-25311R1

Risk factors for 3 - month mortality in bedridden patients with hospital - acquired pneumonia: a multicenter prospective study.

PLOS ONE

Dear Dr. Li,

Thank you for submitting your manuscript to PLOS ONE. After careful consideration, we feel that it has merit but does not fully meet PLOS ONE’s publication criteria as it currently stands. Therefore, we invite you to submit a revised version of the manuscript that addresses the points raised during the review process.

We look forward to receiving your revised manuscript.

Kind regards,

Shane Patman, PhD

Academic Editor

PLOS ONE

Journal Requirements:

Additional Editor Comments (if provided):

For this revision it was fortuitous to continue with the same peer reviewers from the initial submission. The sentiment from this review cycle is predominantly positive. Reviewer 2 notes an opportunity for slight adjustment to the manuscript discussion to enhance clarity and provide direction for future research efforts, as detailed below.

Reviewers' comments:

Reviewer's Responses to Questions

**Comments to the Author**

1. If the authors have adequately addressed your comments raised in a previous round of review and you feel that this manuscript is now acceptable for publication, you may indicate that here to bypass the “Comments to the Author” section, enter your conflict of interest statement in the “Confidential to Editor” section, and submit your "Accept" recommendation.

Reviewer #1: All comments have been addressed

Reviewer #2: All comments have been addressed

2. Is the manuscript technically sound, and do the data support the conclusions?

Reviewer #1: Yes

Reviewer #2: Yes

3. Has the statistical analysis been performed appropriately and rigorously? 

Reviewer #1: Yes

Reviewer #2: Yes

4. Have the authors made all data underlying the findings in their manuscript fully available?

Reviewer #1: Yes

Reviewer #2: Yes

5. Is the manuscript presented in an intelligible fashion and written in standard English?

Reviewer #1: Yes

Reviewer #2: (No Response)

6. Review Comments to the Author

Reviewer #1: (No Response)

Reviewer #2: Excellent revision. All of my major questions were addressed. One minor comment - authors discuss previous severity assessment tools (PSI/CURB-65), but these were intended for use to predict 30-day mortality in patients with community-onset pneumonia - it should probably be highlighted that these tools were never trained or designed for the use in patients with HAP. The lack of severity assessment tools specifically for this group of patients could rather be highlighted.

7. PLOS authors have the option to publish the peer review history of their article (what does this mean?). If published, this will include your full peer review and any attached files.

Reviewer #1: No

Reviewer #2: **Yes: **Barbara Jones

---

## [Author Response · Author response to Decision Letter 1]

11 Mar 2021

Journal Requirements:

Author response: Thank you for the reminder. We have reviewed the reference list and ensured that it is completed and correct.

Additional Editor Comments (if provided):

For this revision it was fortuitous to continue with the same peer reviewers from the initial submission. The sentiment from this review cycle is predominantly positive. Reviewer 2 notes an opportunity for slight adjustment to the manuscript discussion to enhance clarity and provide direction for future research efforts, as detailed below.

Response：Thanks you for your positive comments. We have revised the discussion section according to the Reviewer’s suggestion.

Reviewers' comments:

Reviewer's Responses to Questions

Comments to the Author

1. If the authors have adequately addressed your comments raised in a previous round of review and you feel that this manuscript is now acceptable for publication, you may indicate that here to bypass the “Comments to the Author” section, enter your conflict of interest statement in the “Confidential to Editor” section, and submit your "Accept" recommendation.

Reviewer #1: All comments have been addressed

Reviewer #2: All comments have been addressed

2. Is the manuscript technically sound, and do the data support the conclusions?

Reviewer #1: Yes

Reviewer #2: Yes

3. Has the statistical analysis been performed appropriately and rigorously? 

Reviewer #1: Yes

Reviewer #2: Yes

4. Have the authors made all data underlying the findings in their manuscript fully available?

Reviewer #1: Yes

Reviewer #2: Yes

5. Is the manuscript presented in an intelligible fashion and written in standard English?

Reviewer #1: Yes

Reviewer #2: (No Response)

6. Review Comments to the Author

Reviewer #1: (No Response)

Reviewer #2: Excellent revision. All of my major questions were addressed. One minor comment - authors discuss previous severity assessment tools (PSI/CURB-65), but these were intended for use to predict 30-day mortality in patients with community-onset pneumonia - it should probably be highlighted that these tools were never trained or designed for the use in patients with HAP. The lack of severity assessment tools specifically for this group of patients could rather be highlighted.

Response: Thanks for your comments. Based on your helpful suggestions, we have made the necessary corrections to our previous draft. Detail of the corrections in the discussion is provided below.

With regard to the severity of illness, previous studies have mainly focused on Pneumonia Severity Index (PSI) and CURB-65 score, which were mainly used to predict mortality in patients with CAP [30] and were seldom trained or designed in patients with HAP [31]. The lack of severity assessment tools specifically for this group of patients has limited the ability to early detection of poor prognosis among HAP patients. The current study added new evidence by exploring the role of chronic comorbidities in predicting HAP patients’ outcomes. We found that the ACCI score was an independent risk factor for mortality among HAP patients. The ACCI is widely used to predict mortality and has been validated in various clinical populations [21, 32]. The results support the prognostic value of the ACCI score and indicate that the ACCI score may be useful for risk stratification and decision making of individual HAP patients. Future studies are necessary to compare its performance with CURB-65 and PSI and determine if this tool may be universally applied to HAP patients.

7. PLOS authors have the option to publish the peer review history of their article (what does this mean?). If published, this will include your full peer review and any attached files.

Do you want your identity to be public for this peer review? For information about this choice, including consent withdrawal, please see our Privacy Policy.

Reviewer #1: No

Reviewer #2: Yes: Barbara Jones

---

## [Editor Report · Decision Letter 2]

15 Mar 2021

Risk factors for 3 - month mortality in bedridden patients with hospital - acquired pneumonia: a multicenter prospective study.

PONE-D-20-25311R2

Dear Dr. Li,

We’re pleased to inform you that your manuscript has been judged scientifically suitable for publication and will be formally accepted for publication once it meets all outstanding technical requirements.

Kind regards,

Shane Patman, PhD

Academic Editor

PLOS ONE
---

## [Editor Report · Acceptance letter]

22 Mar 2021

PONE-D-20-25311R2 

Risk factors for 3-month mortality in bedridden patients with hospital-acquired pneumonia: a multicentre prospective study. 

Dear Dr. Li:

I'm pleased to inform you that your manuscript has been deemed suitable for publication in PLOS ONE. Congratulations! Your manuscript is now with our production department. 

Kind regards, 

on behalf of

Assoc Prof Shane Patman 

Academic Editor

PLOS ONE